# The Use of Removed Mesquite Brush as a Fiber Replacement in Silage Production

**DOI:** 10.3390/ani12202795

**Published:** 2022-10-16

**Authors:** William Taylor Fair, Jeffrey Bryan Breeden, Thomas Wayne Atchley, Barry Don Lambert, Zhan Aljoe, Walter Franklin Owsley, William Brandon Smith

**Affiliations:** 1Department of Animal Science, Tarleton State University, Stephenville, TX 76402, USA; 2Department of Wildlife and Natural Resources, Tarleton State University, Stephenville, TX 76402, USA; 3University College, Tarleton State University, Stephenville, TX 76402, USA; 4College of Agriculture and Natural Resources, Tarleton State University, Stephenville, TX 76402, USA; 5Department of Animal Sciences, Auburn University, Auburn, AL 36849, USA

**Keywords:** mesquite, *Prosopis*, silage, inoculant, fermentation length, pretreatment, feedstock, sustainability

## Abstract

**Simple Summary:**

Mesquite is considered an invasive browse species in most of the American Southwest, spreading rapidly into Texas and leading to disruption of native grasslands. Management efforts generate a significant volume of organic debris. Incorporation of this organic material into livestock feeding efforts would represent a sustainable solution to this ecological problem. Our objectives were to (1) determine the nutritive value and fermentation characteristics of silage produced with mesquite biomass, and (2) evaluate potential pre-treatment methods of mesquite biomass prior to ensiling. Inclusion of mesquite biomass in a bermudagrass-based silage was shown to increase fiber concentrations, decrease crude protein and in vitro digestibility, and decrease fermentation quality (pH and volatile fatty acids [VFA]). However, 250 g kg^−1^ inclusion of mesquite was shown to be similar to grass silage alone. There was no effect of lactic acid bacteria inoculation, though increasing length of incubation did increase VFA production and decrease silage pH. Pre-treatment of mesquite biomass with acid or alkali did not improve ensiling properties. Results are interpreted to mean that mesquite biomass may be effectively incorporated into silage at levels up to 250 g kg^−1^.

**Abstract:**

Mesquite (*Prosopis* L.) is considered an invasive browse species in most of the American Southwest. Mechanical intervention removes yields an excess of organic debris. Anecdotal evidence in the past has supported using such browse as feed for livestock. Thus, our objectives were to (1) determine the nutritive value and fermentation characteristics of silage produced with mesquite biomass, and (2) evaluate solvent treatment of mesquite biomass prior to ensiling. In Experiment 1, we evaluated mesquite inclusion rate (0, 250, 500, 750, or 1000 g kg^−1^ DM), length of fermentation (28, 56, or 84 d), and silage inoculant. In Experiment 2, we evaluated the effects of mesquite pre-treatment with acid (H_2_SO_4_) or alkali (NaOH) solutions. Concentrations of NDF, ADF, and ADL, as well as IVTD, decreased (*p* < 0.05) with increasing mesquite inclusion. However, 250 g mesquite kg^−1^ DM did not differ from grass silage. There was no effect (*p* > 0.05) of inoculation, though increasing length of incubation did increase (*p* < 0.05) VFA production and decrease (*p* < 0.05) silage pH. Solvent treatment did not improve ensiling properties. Results are interpreted to mean that mesquite biomass may be effectively incorporated into silage at levels up to 250 g kg^−1^.

## 1. Introduction

Mesquite (*Prosopis* L.) is considered an invasive browse species in most of the American Southwest. Though native to the area, its historic density was considerably smaller than it is in modern times. With increased density in the southwestern United States, there has been a high demand for both the removal of mesquite in fields and grasslands as well as preventing regrowth. Mechanical intervention removes the taproot and lignotuber of the plant from the soil to effectively kill the plant [1]. However, mechanical removal yields an excess of biomass that must be dealt with, typically by burning [2]. This necessitates solutions to invasive plant control that are environmentally sustainable and supportive of productive practices. In times of drought or limited traditional feedstuff production, livestock producers have turned to non-traditional feedstocks, such as tree wood, brush, and high-fiber crop residue, to feed their livestock [3]. Mesquite, in particular, has garnered more recent attention [4,5,6]. Dutta et al. [7] found that leaves of *P. cineraria* (L.) Druce improved the digestibility coefficients of all measures of nutritive value when supplemented to a rice straw diet in goats, while Bhatta et al. [8] found mixed results of *P. cineraria* leaf supplementation of lambs and kids. Much of the interest in feeding mesquite species to livestock has revolved around the use of pods [9,10,11,12] or seeds [13] as a protein source as mesquite belongs to the Fabaceae family [5]. Few, though, have investigated the total biomass of mesquite. Adamu et al. [14] found in a dose-titration study with rabbits that *P. africana* (Guill., Perr. & A. Rich.) Taubert pulp replacement at levels of 200 g kg^−1^ diet resulted in similar or improvement growth performance. However, the limitation of such a feeding strategy is the inherently low digestibility of wood products [3]. Such materials can be treated with strong acids or alkalis to loosen the lignocellulose matrix for digestion by ruminant animals [3,15,16,17,18,19], and the process may be improved with the addition of increased heat and pressure [20]. However, to date there has been no investigation into the incorporation of mesquite biomass into silages for use in livestock feeding. Thus, our objectives were to (1) determine the nutritive value and fermentation characteristics of silage produced with mesquite biomass, and (2) evaluate potential pre-treatment methods of mesquite biomass prior to ensiling.

## 2. Materials and Methods

### 2.1. Experiment 1

In our first experiment, we evaluated the ratio of mesquite to ‘Coastal’ bermudagrass (*Cynodon dactylon* [L.] Pers.), length of fermentation, and use of inoculant in the creation of mesquite-based silage. This experiment was conducted as a randomized complete block design with a 5 × 3 × 2 factorial treatment structure. This experiment was conducted in both the fall and spring (blocks) to encompass both early season and late season growth. Mesquite inclusion rate (factor 1) ranged from 0 to 1000 g kg^−1^ DM in 250 g kg^−1^ increments. Fermentation lengths (factor 2) were 28, 56, and 84 d. Inoculant (factor 3) was a set amount of 1 µL/g DM for all samples that received it.

Coastal bermudagrass was harvested using mechanical hedge clippers and placed into sterilized buckets for transport to the lab. The harvested grass was between 6 and 30 cm in height at the time of collection. Mesquite branches that were 6 cm in diameter were collected from actively growing trees to be processed for the silage production. The branches that were selected mimicked the appearance of a mature tree, with most of the mass being woody and leaves being present on the last one-third of the branch. After collection, the mesquite was passed twice through an 8.8-cm Power King woodchipper. By passing the wood through the woodchipper twice, the pieces were ensured to be 4 × 8 cm in size or smaller. The chips were collected in a sterilized bucket as they left the woodchipper to be transported to the lab. The equivalent of 200 g DM of each sample was placed into 28 × 30 cm vacuum bags with the bags in the inoculant group receiving 1 µL/g DM. Lactic acid bacteria (Biotal Plus II silage inoculant; Lallemand Animal Nutrition, Milwaukee, WI, USA) was used as the inoculant for this experiment. Prior to sealing, distilled water was added to the bags as needed to get bring the moisture content to 650 g kg^−1^. This proportion was chosen as it is the minimum moisture content for traditional silage production [21]. The bags were then vacuum sealed and placed into an incubator that was set to 29 °C for a period of 28, 56, or 84 d. After this period of incubation, the bags were removed and divided into two aliquots. The first aliquot was dried in a forced air oven at 55 °C, ground to pass through a 2-mm or 1-mm screen (dependent on assay) using a Wiley mill, and assayed for NDF and ADF sequentially according to the procedures of Vogel et al. [22] using an ANKOM^200^ Fiber Analyzer. Acid detergent lignin was assayed on the ADF residues according to the procedures of AOAC [23]. In vitro true digestibility was assayed according to the procedures of Vogel et al. [22] using a Daisy^II^ incubator. Crude protein was assayed using the Leco 828 CN Analyzer (Method 990.03 [23]; Leco Corporation, St. Joseph, MO, USA). The second aliquot was immediately frozen and shipped to Cumberland Valley Analytical Service (Waynesboro, PA, USA) for fermentation product and pH analysis.

### 2.2. Experiment 2

In our second experiment, our objective was to evaluate the effects of mesquite pre-treatment with acid or alkali solutions on silage production. This experiment was conducted as a randomized complete block design with a 2 × 4 × 2 + 1 augmented factorial design. This experiment was conducted in both the spring and the fall (blocks) to encompass both early and late season growth. There were two levels of pre-treatment (factor 1; H_2_SO_4_ or NaOH). Each pre-treatment was applied at four concentrations (factor 2; 5, 10, 15, or 20 mL L^−1^ [expressed as 0.25, 0.50, 0.75, or 1.00 N for H_2_SO_4_]). Inoculant (factor 3) was set at 1 µL/g DM for all samples that received it. A single negative control (vegetative material with no pre-treatment) was added onto the factorial. Fermentation length for this experiment was set at 28 d. Prior to sealing distilled water was added to the bags as needed to get bring the moisture content up to 65%.

Mesquite branches were harvested and chipped as described for Experiment 1. The acid used in this experiment (720 mL H_2_SO_4_ L^−1^) was diluted to 0.25, 0.50, 0.75 and 1.00 N with distilled H_2_O to a volume of 800 mL. Sodium hydroxide (NaOH) was prepared at 5, 10, 15, and 20 mL NaOH L^−1^ with distilled H_2_O to a volume of 800 mL. The solutions were created and added to 127.56 g of mesquite (equal to 100 g DM). There were 2 trays made for each treatment combination in each block. Solutions were mixed with mesquite chips and allowed to sit for a period of 3 h. Every hour during this period, the mixtures were stirred to help evenly distribute the chemicals into the wood. After the 3-h period, solutions were drained, and the residue was rinsed with distilled water until it had reached a neutral pH [19]. After rinsing, the wood was prepared for ensiling. One tray from each treatment group received inoculant. All groups received 40 g of H_2_O to bring their DM to acceptable ensiling levels. The bags were vacuum-sealed and placed into an incubator that was set to 29 °C for a period of 28 d. Following incubation, sample preparation and laboratory assays were conducted as described in Experiment 1.

### 2.3. Statistical Analysis

Data for each experiment were analyzed using SAS v. 9.4 (SAS Institute, Inc., Cary, NC, USA). Prior to analysis, raw data were tested using the NORMAL option of PROC UNIVARIATE to ensure data normality. Normality was assumed when Shapiro–Wilk’s *W* met or exceeded 0.9 [24,25].

In Experiment 1, data were analyzed using the generalized linear mixed models procedure (PROC GLIMMIX) of SAS. The fixed effects included mesquite inclusion rate, length of incubation, inoculant, and all two- and three-way interactions. Denominator degrees of freedom were adjusted using the 2nd order Kenward-Roger approximation [26]. The random statement included the effect of block. Means separations were performed based on *F*-protected *t*-tests using the LINES option in the LSMEANS statement of PROC GLIMMIX. *p*-values of mean differences were adjusted using the Tukey–Kramer approximation for small sample sizes [27].

In Experiment 2, data were analyzed using the generalized linear mixed models procedure (PROC GLIMMIX) of SAS. The fixed effects included treatment (a linear combination of pre-treatment solution and concentration) and inoculant. Denominator degrees of freedom were adjusted using the 2nd order Kenward-Roger approximation [26]. The random statement included the effect of block. Means separations for treatment were performed based on *F*-protected *t*-tests using the LINES option in the LSMEANS statement of PROC GLIMMIX. The Dunnett’s test was used to compare each treatment combination to the negative control in the augmented factorial. Means separations for inoculant were performed based on *F*-protected *t*-tests using the LINES option in the LSMEANS statement of PROC GLIMMIX. *p*-values of mean differences among inoculant levels were adjusted using the Tukey–Kramer approximation for small sample sizes [27]. Orthogonal contrasts were used to assess the main effects of solvent type and concentration.

For both experiments, the α-level for mean differences was set at 0.05. When interactions had *p* < α, the interaction was discussed; otherwise, main effects were discussed. To ensure proper analysis, residuals were tested using the NORMAL option of PROC UNIVARIATE, and normality conditions were assumed as previously described.

## 3. Results

### 3.1. Mesquite as Feedstock

Measures of nutritive value from *Prosopis* species used as feedstock are presented in Table 1. At least eight different *Prosopis* species have been evaluated as feedstock: African mesquite (*P. africana*), algarrobo (*P. chilensis* [Molina] Stuntz), jand (*P. cineraria*), tortuous mesquite (*P. flexuosa*), mesquite (*P. juliflora* [Sw.] DC.), smooth mesquite (*P. laevigata* [Humb. & Bonpl. ex Willd.] M.C. Johnst.), kiawe (*P. pallida* [Humb. & Bonpl. ex Willd.] Kunth), and *P. ruscifolia* Griseb. Among these, the most popularly studied species was the common mesquite, *P. juliflora*. Additionally, researchers have evaluated leaves, pods, pulp, and seeds as specific feed ingredients. The most commonly evaluated plant part was the pod. Across the reported values, mesquite averages 825 g DM kg^−1^, 357 g NDF kg^−1^ DM, 245 g ADF kg^−1^ DM, 113 g ADL kg^−1^ DM, and 197 g CP kg^−1^ DM.

### 3.2. Experiment 1

As the replacement of grass with mesquite increased, there was a steady decrease (*p* < 0.05) in the nutritive value and fermentation characteristics of the silage (Table 2). The 1000 g kg^−1^ grass silage had the greatest (*p* < 0.05) CP, IVTD, lactate, total VFA, and ammonia, and the lowest (*p* < 0.05) fiber concentrations and pH. By comparison, the 1000 g kg^−1^ mesquite silage had the lowest (*p* < 0.05) CP, IVTD, lactate, total VFA, and ammonia, and the greatest (*p* < 0.05) fiber concentrations. However, pH followed an inverse pattern whereby the greater (*p* < 0.05) pH was found in grass silage versus mesquite silage. Combinations of grass and mesquite tested as silage presented intermediate (*p* > 0.05) values for NDF, ADF, ADL, IVTD, lactate, total VFA, and ammonia. Notably, however, there was no difference among treatments for acetate (*p* = 0.72) or butyrate (*p* = 0.71) production, two measures of silage quality.

Inoculation of mesquite-and-grass silage with a commercial lactic acid bacteria preparation resulted in no differences (*p* > 0.05) in any of the nutritive value or fermentation characteristics measured (Table 3).

As mesquite (as a woody species) is a more recalcitrant feedstock, we were interested in the effect of incubation time on nutritive value parameters and fermentation characteristics of silage (Table 4). Incubation time had no effect (*p* ≥ 0.06) on NDF, ADF, CP, or IVTD, or on lactate, acetate, butyrate, or ammonia production. However, there was an observed increase (*p* < 0.05) in ADL concentrations when incubation was extended from 56 to 84 d. Similarly, total VFA production increased (*p* < 0.05) and pH decreased (*p* < 0.05) with an extension in incubation time.

### 3.3. Experiment 2

The effect of pre-treatment of mesquite biomass with acid or alkali on nutritive value parameters and fermentation characteristics is presented in Table 5. No treatment differed from the control (*p* > 0.05) according to the Dunnett’s test for NDF, ADL, CP, IVTD, or lactate.

There was an effect of solvent concentration (*p* < 0.05) for CP and all fermentation products (Table 5). There was no defined pattern for CP (40, 37, 39, and 36 g CP kg^−1^ DM from 5, 10, 15, and 20 mL L^−1^ concentrations, respectively). However, fermentation products generally decreased with increasing solvent concentration.

There was an effect of solvent type (H_2_SO_4_ vs. NaOH; *p* < 0.05) for all parameters measured (Table 5). Fiber concentrations were greater (*p* < 0.05) when mesquite was treated with NaOH (788, 634, and 244 g NDF, ADF, and ADL kg^−1^ DM, respectively) versus H_2_SO_4_ (723, 569, 203 g NDF, ADF, and ADL kg^−1^ DM, respectively). Crude protein was greater (*p* < 0.05) from H_2_SO_4_− than NaOH-treated mesquite silage (39 vs. 37 g kg^−1^ DM). Similarly, IVTD was greater (*p* < 0.05) from H_2_SO_4_− than NaOH-treated mesquite silage (306 vs. 268 g kg^−1^ DM). As expected, silage pH was greater (*p* < 0.05) from NaOH- than H_2_SO_4_-treated mesquite silage (5.3 vs. 2.6).

Similar to concentration effects, there was an effect of solvent pre-treatment versus the control (*p* < 0.05) for CP and all fermentation products (Table 5). Silage CP concentration decreased (*p* < 0.05) with solvent pre-treatment (46 vs. 38 g CP kg^−1^ DM). Similarly, lactate, acetate, butyrate, total VFA, and ammonia concentrations were all reduced with solvent pre-treatment (1, 9, 9, 20, and 3 g kg^−1^ DM vs. 0, 4, 3, 7, and 1 g kg^−1^ DM, respectively).

As observed in the first experiment, inoculation of mesquite silage with a commercial lactic acid bacteria preparation resulted in no differences (*p* > 0.05) in any of the nutritive value or fermentation characteristics measured (Table 6).

## 4. Discussion

### 4.1. Mesquite as Feedstock

Many investigators over time have evaluated wood products (raw, treated, and byproducts) for their suitability as feedstuffs for ruminant animals [35]. Due to its invasive nature, especially in the American Southwest [36,37,38], mesquite wood from removal and eradication efforts represents a potentially suitable option for feedstock selection [5,36]. The ensiled mesquite used in Experiment 1 of this study was comparable to reported figures for nutritive value (Table 1). The mesquite silage presented the greatest value for NDF of all documented figures. This is not surprising, however, given that most investigations have selected individual plant parts, such as pods [10,11,12,14,31,32], leaves [7,8,29], or seeds [14,28,33,34]. Pulp, the most similar product to the whole-plant biomass used in the current study, was only evaluated in one previous manuscript [9], and this experiment did not evaluate the commonly found *P. juliflora*. Notably, NDF, ADF, and ADL were among the greatest in reported values, supporting the notion of high recalcitrant fiber components in mesquite whole-plant biomass. Crude protein, the primary nutrient of interest in a leguminous feedstuff such as mesquite, was similar to the values observed from mesquite pods (even though pods were not readily visible in the ensiled material) and pulp, but much less than the reported values for seeds (as would be expected).

In the arid regions where mesquite growth is most observed, small ruminants are the predominant livestock species produced [5]; thus, most evaluations of mesquite feedstock suitability have been conducted with either sheep or goats. Ravikala et al. [32] used mesquite pods as a protein source at up to 300 g kg^−1^ ration in feedlot lambs; there were no differences in any measure of animal performance with *P. juliflora* pod inclusion. Inclusion of *P. juliflora* pods up to 400 g kg^−1^ ration for sheep also resulted in no changes in DM or OM digestibility or ruminal VFA production, though there was a linear decrease in N retention [11]. When these pods were included at a similar rate in the ration of growing goats, however, animal performance (feed intake, weight gain, and feed conversion) was inhibited at inclusion rates above 200 g kg^−1^ [31].

A few researchers have broadened the scope of mesquite feedstock inclusion to address animals other than small ruminants. Feed intake, feed conversion ratio, and weight gain were all decreased when *P. laevigata* seeds were fed in isonitrogenous diets to broiler chicks [13]. Average daily gain of rabbits was not affected by inclusion of *P. africana* pulp, though feeding costs and cost of production were decreased with increasing inclusion [14].

### 4.2. Ensiling Wood Products

Of primary interest in our study was the use of mesquite in a silage product as a potential feedstuff for ruminant animals. Though we found no report of ensiled mesquite in the literature, the ensiling of trees and browse for feedstock is not an entirely unique concept. Phiri et al. [39] found that inclusion of *Acacia boliviana* Rusby and white leadtree (*Leucaena leucocephala* [Lam.] de Wit) increased the CP concentration of maize-tree legume silages without impact on the fiber concentrations, though fiber and protein concentrations both decreased after the ensiling process. Tree legume inclusion in that study also resulted in an increase in acetic acid and ammonia production, though pH was greater than solely maize silage [39]. When woman’s tongue (*Albizia lebbeck* [L.] Benth.) or Florida fishpoison tree (*Piscidia piscipula* [L.] Sarg.) were included in elephant grass (*Pennisetum purpureum* Schumach.) silage, there was no change in DM or OM digestibility or in milk yield or components in lactating goats, but fiber digestibility was increased [40]. In our experiment, mesquite inclusion increased the concentration of NDF, ADF, and ADL of a bermudagrass silage while decreasing CP concentrations, IVTD, and production of lactic acid and total VFA. Though our experiment produced a silage composed solely of tree biomass, previously described studies only evaluated the inclusion of tree species with a more conventional vegetative silage substrate.

### 4.3. Silage Inoculation

Reports in the literature are varied with regard to the effect of inoculation of vegetative material for silage production. Aksu et al. [41] found that the addition of a lactic acid bacteria preparation to maize silage increased DM and NDF digestibility and lactic acid production while decreasing both pH and butyric acid production. Nkosi et al. [42] found that potato (*Solanum tuberosum* L.) hash/wheat (*Triticum aestivum* L.) bran treated with a similar inoculant to the one used in our study resulted in increased CP digestibility and N retention. However, Shoup et al. [43] found that inoculant had little effect on the fermentation of cool-season annual forage mixtures for baleage production. In both experiments of our study, no effect of inoculant was observed for nutritive value parameters or fermentation characteristics.

### 4.4. Length of Incubation

In Experiment 1, we hypothesized that extending the length of incubation would improve fermentation of a recalcitrant plant such as mesquite. Increasing fermentation length from 7 to 112 d resulted in a linear increase in CP concentration and in situ digestibility and a linear decrease in pH in whole plant maize silage [44]. Similarly, high-moisture maize silage stored for up to 300 d resulted in increased DM digestibility, both with or without inoculation [45]. In our experiment, though, there was little change in silage quality when moving from 28 to 84 d of incubation. There was an increase in ADL concentration between 56 and 84 d of incubation, and pH decreased and total VFA production increased with increasing incubation length (Table 4). However, there was no effect of incubation length on IVTD, the more accurate measure of feedstock quality.

### 4.5. Solvent Treatment of Fibrous Feedstock

In light of the resistance of woody and fibrous feedstuffs to digestion, many researchers have investigated solvent treatment of plant material to enhance digestibility. Hellriegel and Lucanus [15] attempted to increase the feeding value of straw by mechanical processing, water addition, and heat, but this failed to improve digestibility. However, addition of NaOH was shown in early experiments with straw to increase starch and sugar concentrations without negatively impacting intake by sheep [17]. Ellenberger [46] reported that hydrolyzed wood meal represented a viable feedstuff when used for working horses. Nowicka et al. [47] found that acid hydrolysis of maize silage resulted in an increases release of sugar substrates for biogas production, which could be viewed as a surrogate for digestibility. In Experiment 2, while there was an effect of solvent pre-treatment (either acid or alkali) on chemical composition and fermentation, the results do not indicate an improvement in silage quality relative to the untreated control (Table 5).

## 5. Conclusions

In conclusion, we believe that whole-plant biomass from mesquite may be effectively incorporated into silage production for livestock feeding. If we consider 500 g IVTD kg^−1^ DM to be to lowest acceptable value for production of beef cattle [48], then a 500 g kg^−1^ rate of mesquite inclusion for silage would be an acceptable mixture (Table 2). However, since the 250 g kg^−1^ mesquite inclusion rate silage did not differ from the grass silage in IVTD or total VFA or ammonia concentrations, this would likely be the more suitable choice for application. There is no indication that inoculation with lactic acid bacteria or pre-treatment with acid or alkali provides any additional benefit to the ensiling process. Results show promise for future efforts to incorporate a mesquite-based silage into ruminant feeding trials.

## Figures and Tables

**Table 1 animals-12-02795-t001:** Nutritive value of mesquite (*Prosopis* spp.) reported in the literature as feedstock.

Species	Plant Part	DM *	NDF	ADF	ADL	CP	Tannins	Sources
*P. africana*	Pods	760	-	-	-	185	-	[14]
*P. africana*	Pulp	860	-	-	-	100	57.5	[14]
*P. africana*	Seeds	850	-	-	-	277	-	[14]
*P. chilensis*	Seeds	-	-	-	-	254	-	[28]
*P. cineraria*	Leaves	-	446	352	-	155	70.0	[7]
*P. cineraria*	Leaves	502	567	360	189	159	90.7	[8]
*P. flexuosa*	Leaves/twigs	-	397	-	-	141	-	[29]
*P. juliflora*	Leaves	923	271	182	-	216	-	[30]
*P. juliflora*	Pods	-	391	276	36	225	-	[11]
*P. juliflora*	Pods	930	402	317	-	120	-	[31]
*P. juliflora*	Pods	-	-	-	-	134	-	[32]
*P. juliflora*	Pod meal	944	246	-	-	94	-	[12]
*P. juliflora*	Seed meal	902	-	-	-	330	8.3	[33]
*P. juliflora*	Silage	-	686	519	174	99	-	this study
*P. laevigata*	Pods	917	265	169	-	117	-	[10]
*P. laevigata*	Roasted pods	955	259	182	-	123	-	[10]
*P. laevigata*	Seeds	925	329	118	-	394	-	[13]
*P. pallida*	Pulp	437	-	-	-	401	-	[9]
*P. ruscifolia*	Seeds	-	-	-	-	129	-	[5,34]

* DM, g kg^−1^ = dry matter; NDF, g kg^−1^ DM = neutral detergent fiber; ADF, g kg^−1^ DM = acid detergent fiber; ADL, g kg^−1^ DM = acid detergent lignin; CP, g kg^−1^ DM = crude protein.

**Table 2 animals-12-02795-t002:** Effect of mesquite inclusion rate on nutritive value components and fermentation characteristics of silage used in the evaluation of mesquite wood inclusion.

	Mesquite Inclusion Rate, g kg^−1^ DM							Contrasts	
Response *^,†^	1000	750	500	250	0	SEM ^‡^	*p*-Value ^§^	L	Q
NDF	686 ^a^	675 ^ab^	647 ^ab^	606 ^ab^	592 ^b^	60.8	0.01	0.06	0.89
ADF	519 ^a^	465 ^ab^	412 ^bc^	352 ^cd^	301 ^d^	20.7	<0.01	<0.01	0.07
ADL	174 ^a^	129 ^b^	102 ^c^	71 ^d^	39 ^e^	10.5	<0.01	<0.01	<0.01
CP	99 ^b^	107 ^b^	131 ^b^	152 ^a^	174 ^a^	33.7	<0.01	<0.01	0.50
IVTD	382 ^d^	441 ^cd^	512 ^bc^	588 ^ab^	665 ^a^	62.7	<0.01	<0.01	0.13
pH	4.5 ^b^	4.7 ^ab^	4.8 ^a^	4.9 ^a^	4.9 ^a^	0.14	<0.01	0.82	0.30
Lactate	5 ^c^	14 ^bc^	21 ^bc^	29 ^ab^	39 ^a^	13.4	<0.01	<0.01	0.11
Acetate	12	9	10	11	12	2.5	0.72	0.18	0.68
Butyrate	8	6	9	7	5	5.3	0.71	0.62	0.37
Total VFA	24 ^d^	29 ^cd^	41 ^bc^	49 ^ab^	58 ^a^	5.5	<0.01	<0.01	0.38
Ammonia	9 ^d^	13 ^cd^	19 ^bc^	26 ^ab^	27 ^a^	1.6	<0.01	<0.01	0.80

* NDF, g kg^−1^ DM = neutral detergent fiber, assayed with sodium sulfite and heat-stable α-amylase and expressed inclusive of residual ash; ADF, g kg^−1^ DM = acid detergent fiber, expressed inclusive of residual ash; ADL, g kg^−1^ DM = acid detergent fiber, expressed inclusive of residual ash; CP, g kg^−1^ DM = crude protein; IVTD, g kg^−1^ DM = in vitro true digestibility. ^†^ Fermentation products are expressed as g kg^−1^ DM. ^‡^ SEM = standard error of the mean. ^§^ *p*-values were adjusted using the Tukey–Kramer adjustment. Orthogonal contrasts: L = linear; Q = quadratic. ^a–e^ Means within a row without common superscript letters are different (*p* < 0.05).

**Table 3 animals-12-02795-t003:** Effect of inoculant on nutritive value components and fermentation characteristics of silage used in the evaluation of mesquite wood inclusion.

	Inoculated			
Response *^,†^	No	Yes	SEM ^‡^	*p*-Value ^§^
NDF	646	637	58.6	0.61
ADF	415	404	23.6	0.39
ADL	107	99	9.9	0.05
CP	133	133	32.7	0.97
IVTD	516	519	59.8	0.87
pH	4.8	4.7	0.13	0.17
Lactate	21	22	13.0	0.77
Acetate	9	12	3.2	0.17
Butyrate	6	8	4.9	0.54
Total VFA	39	41	3.1	0.64
Ammonia	19	19	1.0	0.92

* NDF, g kg^−1^ DM = neutral detergent fiber, assayed with sodium sulfite and heat-stable α-amylase and expressed inclusive of residual ash; ADF, g kg^−1^ DM = acid detergent fiber, expressed inclusive of residual ash; ADL, g kg^−1^ DM = acid detergent fiber, expressed inclusive of residual ash; CP, g kg^−1^ DM = crude protein; IVTD, g kg^−1^ DM = in vitro true digestibility. ^†^ Fermentation products are expressed as g kg^−1^ DM. ^‡^ SEM = standard error of the mean. ^§^ *p*-values were adjusted using the Tukey–Kramer adjustment.

**Table 4 animals-12-02795-t004:** Effect of length of fermentation on nutritive value components and fermentation characteristics of silage used in the evaluation of mesquite wood inclusion.

	Length of Incubation, d					Contrasts	
Response *^,†^	28	56	84	SEM ^‡^	*p*-Value ^§^	L	Q
NDF	661	636	627	59.4	0.31	0.14	0.69
ADF	417	398	414	24.5	0.47	0.84	0.23
ADL	93 ^b^	98 ^b^	119 ^a^	10.1	<0.01	<0.01	0.10
CP	131	132	136	33.0	0.91	0.68	0.88
IVTD	509	510	533	60.8	0.62	0.39	0.64
pH	4.9 ^a^	4.7 ^ab^	4.6 ^b^	0.13	0.01	<0.01	0.72
Lactate	16	25	25	13.1	0.06	0.04	0.27
Acetate	11	9	12	3.3	0.36	0.77	0.16
Butyrate	5	6	9	5.1	0.29	0.15	0.51
Total VFA	33 ^b^	41 ^ab^	47 ^a^	5.0	0.01	<0.01	0.89
Ammonia	18	17	21	1.2	0.11	0.16	0.11

* NDF, g kg^−1^ DM = neutral detergent fiber, assayed with sodium sulfite and heat-stable α-amylase and expressed inclusive of residual ash; ADF, g kg^−1^ DM = acid detergent fiber, expressed inclusive of residual ash; ADL, g kg^−1^ DM = acid detergent fiber, expressed inclusive of residual ash; CP, g kg^−1^ DM = crude protein; IVTD, g kg^−1^ DM = in vitro true digestibility. ^†^ Fermentation products are expressed as g kg^−1^ DM. ^‡^ SEM = standard error of the mean. ^§^ *p*-values were adjusted using the Tukey–Kramer adjustment. Orthogonal contrasts: L = linear; Q = quadratic ^a,b^ Means within a row without common superscript letters are different (*p* < 0.05).

**Table 5 animals-12-02795-t005:** Effect of solvent pre-treatment on nutritive value components and fermentation characteristics of silage used in the evaluation of pre-treated mesquite wood inclusion.

		Sulfuric Acid, N				Sodium Hydroxide, mL L^−1^				Contrasts					
Response ^†,‡^	CON ^§^	0.25	0.50	0.75	1.00	5	10	15	20	SEM ^#^	CONC	TYPE	CONT	L	Q
NDF	760	694	745	730	724	780	792	788	792	20.7	0.65	<0.01	0.80	0.56	0.89
ADF	601	524 *	587	581	584	625	639	640	633	22.6	0.36	<0.01	0.96	0.19	0.90
ADL	219	191	209	204	207	257	243	243	233	19.1	0.97	<0.01	0.64	0.89	0.48
CP	46	37	37	40	41	42	37	38	30 *	4.4	<0.01	<0.01	<0.01	<0.01	0.28
IVTD	303	293	313	308	308	264	256	260	290	21.8	0.73	0.03	0.33	0.74	0.23
pH	4.1	2.9 *	2.5 *	2.4 *	2.7 *	4.8 *	5.1 *	5.3 *	6.1 *	0.32	0.16	<0.01	0.58	0.18	0.02
Lactate	1	0	0	0	0	0	0	0	0	0.5	0.02	<0.01	<0.01	0.12	0.05
Acetate	9	1 *	2 *	3 *	6	9	4	4 *	2 *	1.3	<0.01	<0.01	<0.01	0.01	<0.01
Butyrate	9	0 *	0 *	0 *	3	9	6	4	3 *	1.8	<0.01	<0.01	<0.01	<0.01	0.07
Total VFA	20	1 *	2 *	3 *	9	20	11	9	3 *	3.1	<0.01	<0.01	<0.01	<0.01	0.02
Ammonia	3	1 *	1 *	1 *	2	2	1 *	1*	1 *	0.3	<0.01	<0.01	<0.01	<0.01	<0.01

^†^ NDF, g kg^−1^ DM = neutral detergent fiber, assayed with sodium sulfite and heat-stable α-amylase and expressed inclusive of residual ash; ADF, g kg^−1^ DM = acid detergent fiber, expressed inclusive of residual ash; ADL, g kg^−1^ DM = acid detergent fiber, expressed inclusive of residual ash; CP, g kg^−1^ DM = crude protein; IVTD, g kg^−1^ DM = in vitro true digestibility. ^‡^ Fermentation products are expressed as g kg^−1^ DM. ^§^ CON = negative control. ^#^ SEM = standard error of the mean. Pre-planned orthogonal contrasts: CONC = comparison among solvent concentrations averaged across solvent types; TYPE = comparison between solvent types averaged across solvent concentrations; CONT = comparison of treatments, averaged across solvent types and concentrations, against the negative control; L = linear effect of solvent concentration; Q = quadratic effect of solvent concentration. * Means within a row differ (*p* < 0.05) from the negative control according to Dunnett’s test.

**Table 6 animals-12-02795-t006:** Effect of inoculant on nutritive value components and fermentation characteristics of silage used in the evaluation of pre-treated mesquite wood inclusion.

	Inoculated			
Response *^,†^	No	Yes	SEM ^‡^	*p*-Value ^§^
NDF	754	758	9.4	0.81
ADF	595	608	12.6	0.37
ADL	219	227	16.2	0.27
CP	40	38	11.6	0.20
IVTD	284	293	11.6	0.49
pH	4.1	3.9	0.22	0.29
Lactate	0.2	0.2	0.21	0.86
Acetate	4.3	4.7	0.61	0.69
Butyrate	3.6	3.4	0.84	0.86
Total VFA	9.1	8.5	1.41	0.73
Ammonia	1.6	1.4	0.12	0.16

* NDF, g kg^−1^ DM = neutral detergent fiber, assayed with sodium sulfite and heat-stable α-amylase and expressed inclusive of residual ash; ADF, g kg^−1^ DM = acid detergent fiber, expressed inclusive of residual ash; ADL, g kg^−1^ DM = acid detergent fiber, expressed inclusive of residual ash; CP, g kg^−1^ DM = crude protein; IVTD, g kg^−1^ DM = in vitro true digestibility. ^†^ Fermentation products are expressed as g kg^−1^ DM. ^‡^ SEM = standard error of the mean. ^§^ *p*-values were adjusted using the Tukey–Kramer adjustment.

## Data Availability

Data summaries are provided in the text of this manuscript. Raw data may be available on request from the corresponding author.

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
