# Peer review of "The Use of Removed Mesquite Brush as a Fiber Replacement in Silage Production"

_animals, 2022, doi:10.3390/ani12202795_

Round 1
Reviewer 1 Report
Review for Manuscript ID: animals-1939481
Summary: This manuscript describes the potential use of mesquite as a feedstuff for ruminants, specifically when processed for silage. This appears to be the first experiment where mesquite of any variety has been processed as a whole shrub and fermented as silage, as most previous work has evaluated feeding different parts of the plant such as leaves, stems, or pulp. In this experiment, the researchers found a novel approach to a very real-world problem in many areas of the American Southwest and evaluated its inclusion with a common feedstuff (bermudagrass silage) found in the area. One major contribution and strength of this project is that this otherwise waste material could be possibly utilized in a productive manner by ruminant animal producers.
General comments about the article: In the experiments themselves, what was the rational for utilizing 28 days of fermentation time for Experiment # 2? Also, were the mesquite materials utilized in both experiments taken at different times or from the same shrubs? There seems to be a big discrepancy between the nutrient analysis of mesquite in Experiment 1 & Experiment 2. Though I am not sure how many previous studies have evaluated it, I believe it would have been helpful to measure the tannin and possibly 5-HTP or serotonin content of the mesquite biomass to detect possible interactions or limitations for feeding to livestock, which would likely be the next logical step in this line of research. What was the negative control utilized for Experiment # 2? Is there any thought on why you observed a pH drop in the mesquite silage vs. grass silage?
General comments about the review: I commend you on being able to find as many references as you were able to regarding using mesquite as a feedstock.
Specific comments: On line 29 in the abstract, this sentence needs to be reworked. In Table 1, for the P. flexuosa those authors did not report a DM value (OMD is presented) and they reported a total tannin (TT) value which is not listed in your table. Also in Table 1, for your study, the values presented appear to be the 0 g kg-1 values which would be the grass silage only values correct? On line 217 & 218, the last sentence does not make sense, as looking at Table 4 total VFA increases but pH decreases with the extension of incubation time, unless the values in Table 4 are backwards. More of a question, but do you have an idea why ADL dropped so much more from d 56 to 84 compared to d 28 to 56? For Table 5, the CP & VFA data seems so much lower compared to the data for Experiment 1; do you have any reason for this discrepancy? For the sentences beginning on line 287 and ending on line 293, these may need to reviewed based upon previous comments regarding Table 1 data for your study compared to the previously published data.
- What was the rational for utilizing 28 days of fermentation time for Experiment # 2?
-Were the mesquite materials utilized in both experiments taken at different times or from the same shrubs? There seems to be a big discrepancy between the nutrient analysis of mesquite in Experiment 1 & Experiment 2.
- I believe it would have been helpful to measure the tannin and possibly 5-HTP or serotonin content of the mesquite biomass to detect possible interactions or limitations for feeding to livestock, which would likely be the next logical step in this line of research.
- What was the negative control utilized for Experiment # 2?
- Is there any thought on why you observed a pH drop in the mesquite silage vs. grass silage?
Author Response
The authors would like to thank the reviewer, especially for the kind words regarding the study and manuscript, but also for the thorough review. We have endeavored to address all of the comments in this response.
General Comments
The rationale for using a 28-d incubation in Experiment 2 was, by convention, this is the minimum amount of time needed to achieve fermentation of ensiled substrates. Though there were statistically significant changes in material ensiled 28, 56, or 84 d in Experiment 1, we did not feel that the changes in fermentation were drastic enough to warrant an extension of the incubation time for testing the second hypothesis.
The mesquite utilized in Experiments 1 and 2 were harvested from the same pasture, but the harvests were conducted on different trees and different days. That would account for the discrepancy in the basal values of the mesquite and is the reason we did not attempt to make comparisons or draw conclusions across experiments.
While we agree that a measure of tannins would be beneficial for telling the story of using mesquite as a feedstock, this was beyond the scope of the current experiments. We also have preliminary data from a separate experiment conducted at this location to suggest that tannins (or, at least, anthocyanins) are destroyed in the ensiling process, thus negating the usefulness in this silage-based manuscript.
The negative control used in Experiment 2 was the same vegetative material used in the other treatments, but it did not undergo any pre-treatment. This has been clarified in L119 in the revised manuscript.
Interpretation of the pH drop in mesquite relative to grass silage would, again, be speculation. One may assume that the green material in the mesquite (leaflets and petioles) may have a higher concentration of cell soluble material relative to a grass which would lend itself to rapid fermentation. However, there is not a concomitant increase in lactic or acetic acid to support that hypothesis. The other speculative hypothesis (not tested in this experiment) could be that the intracellular pH of the mesquite vegetative material is more acidic to begin with than grasses.
Specific Comments
L29 This sentence has been reworded to better reflect the intended statement.
Table 1. The DM value from Allegretti et al. (2012) has been removed. After review of the original document, the reviewer is correct that this value reflected OMD, not DM.
Table 1. The reviewer is correct, and the line item referring to the present study has been corrected. The wrong column was transposed in the original draft.
L225. The reviewer is correct. The statement incorrectly lumped total VFA and pH patterns together when they are, in fact, inverse. This statement has been corrected to accurately reflect the data.
Table 4. The reviewer poses an interesting question. The change in ADL from d 28 to d 56 was slight, but there was a precipitous bump in ADL from d 56 to d 84. I do not know that we have an answer to this question. Our speculation would be that the increase was a proportional response to the removal of other chemical constituents that had been fermented out, but this would be pure speculation and is not fleshed out in any of the data.
L287. This sentence has been updated to accurately reflect the corrected mistake from Table 1.
Reviewer 2 Report
This paper contains interesting results but there are any measurements on animals.
In my opinion, it seems strange to publish it in "Animals".
The main question is the feasability to include ruminants with mesquite brush Except that, I think the manuscript is well written, the tables are clear, the discussion and conclusion in relation with the results and the references appropriate.
Author Response
The authors would like to thank the reviewer for their time and comments. While it may seem that the manuscript may not be a fit for Animals, this special issue on new and potential feedstuffs provides an opportunity to characterize a potential feedstuff in an introductory experiment that can then be built upon by others in performing live animal evaluations.